# The Use of Artificial Intelligence in the Diagnosis and Classification of Thyroid Nodules: An Update

**DOI:** 10.3390/cancers15030708

**Published:** 2023-01-24

**Authors:** Maksymilian Ludwig, Bartłomiej Ludwig, Agnieszka Mikuła, Szymon Biernat, Jerzy Rudnicki, Krzysztof Kaliszewski

**Affiliations:** Department of General, Minimally Invasive and Endocrine Surgery, Wroclaw Medical University, 50-556 Wroclaw, Poland

**Keywords:** artificial intelligence, thyroid nodules, thyroid cancer, diagnosis, ultrasonography, cytopathology, frozen sections, classification, machine learning

## Abstract

**Simple Summary:**

The amount of diagnosed thyroid nodules increases every year. Many researchers have tried to optimize the process of classifying and diagnosing thyroid nodules using artificial intelligence. The aim of this study was to assess the latest applications of artificial intelligence in diagnosing and classifying thyroid nodules. The focus was on innovations in the use of artificial intelligence in the field of ultrasonography and microscopic diagnosis, although other applications were reviewed as well. In total, we analyzed 930 papers published from 2018 to 2022.

**Abstract:**

The incidence of thyroid nodules diagnosed is increasing every year, leading to a greater risk of unnecessary procedures being performed or wrong diagnoses being made. In our paper, we present the latest knowledge on the use of artificial intelligence in diagnosing and classifying thyroid nodules. We particularly focus on the usefulness of artificial intelligence in ultrasonography for the diagnosis and characterization of pathology, as these are the two most developed fields. In our search of the latest innovations, we reviewed only the latest publications of specific types published from 2018 to 2022. We analyzed 930 papers in total, from which we selected 33 that were the most relevant to the topic of our work. In conclusion, there is great scope for the use of artificial intelligence in future thyroid nodule classification and diagnosis. In addition to the most typical uses of artificial intelligence in cancer differentiation, we identified several other novel applications of artificial intelligence during our review.

## 1. Introduction

Thyroid nodules are a common problem encountered in clinical practice. Based on palpation, they have generally been detected in 4–7% of the population; however, due to the current high quality of ultrasound equipment, they are now being diagnosed in as much as 50–70% of the general population [1]. Thyroid cancer is the most common malignancy of the endocrine system [2]. In recent years there has been an upward trend in the detection of this cancer, even to the point of an epidemic, though this may be due in part to overdiagnosis [2,3]. This trend depends mainly on the increased detection of papillary thyroid carcinoma (PTC), while the incidence rates for other thyroid carcinomas, i.e., follicular, medullary, and anaplastic, have remained relatively constant [2,4]. 

When a thyroid nodule is detected, accurate evaluation, classification, and estimation of the risk of malignancy is the most critical issue for recovery, followed by correct therapeutic management. The American Thyroid Association 2015 guidelines recommend that nodules larger than 1 cm should be evaluated. Smaller nodules, i.e., those less than 1 cm in diameter, should be evaluated in special cases, such as when there are other clinical indications. Key diagnostic tools include clinical examination, ultrasonography (US), serum thyrotropin measurement, and fine needle aspiration biopsy (FNAB) [5].

Attempts to use artificial intelligence (AI) in medicine represent a relatively new area of interest. The very idea of creating a machine able to simulate critical thinking appeared for the first time in 1950. The first major “intelligent” computer programs in medicine were MYCIN (1972) and CASNET (1976). MYCIN was designed to generate a list of predicted bacterial pathogens and then select the appropriate antibiotic therapy for a patient on the basis of their weight. CASNET served as a consultation model for glaucoma. For a long period of time, this area developed relatively slowly. Significant development of AI in medicine has only occurred over the past two decades, with the U.S. Food and Drug Administration approving a cloud-based deep learning (DL) application for the first time in 2017 [6]. Currently, the usefulness of AI is being explored in such diverse medical fields as gastroenterology, radiology, oncology, cardiology, ophthalmology, and surgery, with promising results [6,7,8,9,10,11,12,13,14,15]. 

The goal of this paper is to explore the current uses and prospects of AI in diagnosing and classifying thyroid nodules. Special attention is paid to the possibility of using AI in the following areas: ultrasonography, cytopathology, whole-slide imaging of frozen sections, probe electrospray ionization tandem mass spectrometry (PESI-MS), and nuclear medicine, as the above areas seem to be the most interesting and future-orientated. 

## 2. Methods

We searched the Google Scholar and PubMed online databases for studies published from 2018 to 29 October 2022 using mostly mixtures and various forms of the following terms: “artificial intelligence”, “thyroid nodules”, “diagnosis”, “thyroid cancer”, “ultrasonography”, “frozen section”, “cytology”, “nuclear medicine”, “machine learning”, and “pathology”. Three authors went through the databases and decided which articles to choose. Each article was read and confirmed by two authors. In case of a disagreement, the third author was asked to decide whether to reject or accept the article. After analyzing 930 papers and reading the abstract/content of 168 of these, we selected 33 papers that were most related to our topic. We exercised discretion in not selecting multiple works with similar coverage of certain narrow topics in order to cover more areas and assess the diversity of artificial intelligence usage rather than focusing on a single topic. We excluded articles that were not related to the use of artificial intelligence as well as those that did not deal with thyroid nodules. We included both original and review articles. We especially focused on the use of AI in the diagnosis and characterization of pathology based on imaging, such as through ultrasound (US) and microscopy. 

## 3. AI in the Diagnosis and Classification of Thyroid Nodules 

### 3.1. Ultrasonography

In any patient with a suspected thyroid nodule, ultrasonography should be performed as a primary diagnostic tool. The lesion should be evaluated in terms of size, location, and sonographic characteristics (composition, echogenicity, margins, calcifications, shape, and vascularity) [5,16]. It is crucial to properly assess the image seen on the US for further management, as it is this assessment that mainly determines the decision about whether or not to biopsy the lesion when performing further diagnostics [5]. To this end, numerous risk stratification systems have been created to facilitate and standardize decisions; among the most popular are the American Association of Clinical Endocrinologists (AACE)/American College of Endocrinology (ACE)/Associazione Medici Endocrinologi (AME), American College of Radiology Thyroid Imaging Reporting and Data System (ACR TI-RADS), American Thyroid Association (ATA), European Thyroid Association Thyroid Imaging Reporting and Data System (EU-TIRADS), and Korean Society of Thyroid Radiology Thyroid Imaging Reporting and Data System (K-TIRADS). The greatest sensitivity is characterized by ATA (87%) and K-TIRADS (86%), and the highest specificity by ACR TI-RADS (64%). In a head-to-head analysis, ACR TI-RADS has the highest diagnostic odds ratio: ACR TI-RADS vs. ATA at 5.6 vs. 2.9 with *p* = 0.002 and ACR TI-RADS vs. K-TIRADS at 4.5 vs. 2.5 with *p* = 0.002 [17]. However, the above classification systems, which can be referred to as conventional, do not cover all possible features for analysis, such as vascularity and tissue elasticity. Therefore, to achieve even greater accuracy in the evaluation of thyroid nodules and to appropriately categorize the visualized lesion, multimodal scoring systems were created, e.g., French TIRADS and Thyroid Multimodal Imaging Comprehensive (TMC) risk stratification systems, which analyzed even more features included in the US image and showed greater diagnostic performance compared with conventional systems [18]. Suspicious features seen on the image include solid composition, hypoechoic echogenicity, taller-than-wide shape, irregular margins, and calcifications [5,19,20,21]. In their evaluation, multimodal scoring systems additionally use the presence of suspicious lymph nodes, high stiffness on elastography, vascularity, halo effect, comet tail artifact, and a negative score for benign features [18,22].

Algorithms used in research on the usefulness of AI in image evaluation can vary widely, and in clinical practice some achieve better results than others. To better understand the nuances between different algorithms and their results, we have decided to prepare a short introduction, presented below.

Computer programs can essentially be distinguished according to their learning approach as supervised learning and unsupervised learning [23]. 

In supervised learning, a labeled dataset of images already classified by a team of human experts is introduced into an algorithm’s resources. On the basis of these resources, the algorithm tries to independently identify a function that most accurately classifies the provided dataset, the aim being to obtain results as close as possible to the classification made by the human experts. Then, the program compares its results with those contained in labeled datasets. The algorithm then repeats the steps to develop a function with the highest possible accuracy. This is the most common learning approach in medicine [23,24]. An example of such a dataset might include images of pulmonary nodules obtained from computed tomography and classified as cancerous or benign [25]. 

Similarly, in unsupervised learning the algorithm is provided with datasets composed of images, with the difference being that these images are unlabeled. The program learns through its own analysis based on identifying differences and similarities between the inputted images. In this case, there are no reference values made by a human expert for the program to refer to [23,24].

Of the two concepts mentioned above, supervised learning is much more commonly chosen in medicine. It includes several different algorithms, essentially belonging to two different groups: convolutional neural networks (CNNs) and non-neural networks [24]. Deep neural networks, which include CNNs, are considered the gold standard in image analysis [24,26]. CNNs that analyze images are designed along the lines of the animal visual cortex. They are highly complex systems consisting of many different interconnected layers. They extract specific image features and share the features among themselves, analyzing them and ultimately drawing conclusions, which they present as output [26,27,28]. In clinical practice, the key difference between CNNs and non-neural networks, e.g., random forest and support vector machines (SVMs), is that CNNs do not require prior handcrafted feature extraction from the input data. In addition, prior segmentation of tumors by human is not needed for analysis using CNNs [24,27].

The application of AI in the diagnosis of thyroid nodules is developing in many different directions. Programs using both CNN and non-neural methods are being analyzed. The usefulness of the algorithms in making independent decisions and as an adjunct to physicians for even greater accuracy in classifying visualized lesions is being tested.

The most important report from studies conducted in recent years seems to be that AI is on par with experienced radiologists in its assessment and can be successfully used as a computer-aided diagnosis (CAD) system in assisting clinicians, especially less experienced ones, in making further diagnostic and therapeutic decisions (Table 1) [29,30,31,32,33,34,35,36].

He et al. used a CNN (Visual Geometry Group Net) trained on a dataset containing 1421 samples to evaluate 469 nodules in 426 patients. The main limitation of this study is that PTCs accounted for the majority of the malignant nodules. He et al. compared the results to the evaluations of senior radiologists (>10 years of experience) and junior radiologists (<3 years of experience) using ACR TI-RADS. Individually, the junior radiologists scored lower on sensitivity, specificity, and accuracy (63.8%, 83.7%, and 77.6%, respectively) than senior radiologists (81.9%, 88.3%, and 86.4%, respectively) and AI (74.3%, 88.6%, and 84.2%, respectively). Individually, the senior radiologists achieved the highest sensitivity and AI the highest specificity. However, when a junior radiologist was assisted by AI in evaluation, the sensitivity significantly increased to 86.8% and accuracy to 80.4%, although, interestingly, the specificity decreased to 77.5%. Comparison of the receiver operating characteristic (ROC) curve and area under the ROC curve (AUC) showed that junior radiologists had a lower AUC than senior radiologists and AI (0.738, 0.851, and 0.816, respectively). With the assistance of AI, the AUC value of junior radiologists increased to 0.867. There was a statistically significant difference in the AUC values between AI-assisted junior radiologists and junior radiologists (0.867 vs. 0.738; *p* < 0.05). Differences between the AUC values of AI, senior radiologists, and AI-assisted junior radiologists were not significant (*p* > 0.05) [29].

Peng et al. used a CNN (ThyNet) trained on a large dataset of 18,049 images, which is an undoubted advantage of this study. Radiologists were divided into two groups, senior (>8 years of experience) and junior (1–3 years of experience), and evaluated on the basis of the ACR TI-RADS. Three study tests with their own set of samples were assessed: A, B, and C. In Test A, a set of 2185 images of 1424 patients was separately evaluated by AI and radiologists. The CNN proved to be the best according to each performance parameter (sensitivity, specificity, accuracy, and AUC, with 94.9%, 81.2%, 89.1%, and 0.944, respectively). Senior radiologists performed slightly worse than CNN (90.4%, 80.6%, 86.3%, and 0.855), and junior radiologists performed the worst (88.5%, 75.3%, 82.8%, and 0.819). In Test B, 1745 images of 1048 patients were evaluated using the CNN-assisted strategy. AI assistance improved performance in both the senior and junior radiologist groups. It is notable that junior radiologists with CNN assistance achieved better performance than senior radiologists without CNN assistance (sensitivity of 92.4% vs. 90.4%, specificity of 80.8% vs. 80.6%, accuracy of 87.4% vs. 86.3%, and AUC of 86.6% vs. 85.5%, respectively). In Test C, the cooperation of radiologists with CNNs was tested, and videos were additionally evaluated. The sample included 366 images and videos of 303 patients. For radiologists, the total AUC based on the analysis of static images alone was 0.823. When dynamic videos were included, the AUC increased to 0.862. With the further addition of CNN assistance, the AUC was 0.873. Performance was improved in both the senior (*p* = 0.0075) and junior (*p* < 0.0001) radiologist groups. Finally, a scenario was simulated in which biopsy was waived using AI and ACR TI-RADS under strict conditions and the diagnosis was made on the basis of imaging data. A significant reduction in patient FNAB was achieved, from 61.9% to 35.2%, while the missed malignancy rate improved from 18.9% to 17.0% [30].

The potential of deep neural networks to prevent unnecessary FNAB was investigated by Song at al. In their study, the popular Inception-v3 model was used and the K-TIRADS system was used to evaluate tumor malignancy. The study focused on the performance of AI on the basis of the ratio of malignant to benign tumors in the study pool [34], and the results were compared to the work of radiologists in the Chang et al. study [37]. The effectiveness of the Inception-v3 model matched that of the radiologists. The negative predictive value (NPV) of the AI increased as the percentage of malignant samples in the study decreased (90.3%, 93.3%, and 100% NPV when the percentage of malignant samples was 50%, 30%, and 10%, respectively). The sensitivity was variable, but consistently in the 90.9–100.0% range [34]. Both papers (Peng et al. and Song et al.) indicate that AI can be used successfully in the clinician’s daily work, both in the assessment of the malignancy of thyroid nodules and in deciding whether or not to perform FNAB. By relying on computer algorithms, the clinician is able to make decisions appropriate for the therapeutic process without statistically greater harm to the patient than that could potentially result from the evaluation of the US image of the lesion made by a diagnostician without the support of modern algorithms. In addition, the benefit to the patient of not performing FNAB is the avoidance of potential complications that could occur when performing this invasive procedure [30,34].

Kim et al. used S-Detect software, owned by Samsung Medison Co. Ltd., which is a CAD system in two versions, one based on SVMs and one based on CNNs. The performance of both AIs was compared with that of an experienced radiologist with eleven years of experience. In all, 218 thyroid nodules from 106 patients were evaluated. A relatively large sample was represented by malignant nodules, what could have influenced the diagnostic performance of the CAD system. The radiologist had the highest values in terms of sensitivity, specificity, and accuracy (84.9%, 96.2%, and 91.7%, respectively). S-Detect 2 had better sensitivity than S-Detect 1 (81.4% vs. 80.2%, respectively), but lower specificity (68.2% vs. 82.6%, respectively) and accuracy (73.4% vs. 81.7%, respectively). Differences in specificity and accuracy were statistically significant (S-Detect 1 and S-Detect 2 vs. radiologists in both categories *p* < 0.001; S-Detect 1 vs. S-Detect 2, *p* = 0.004 for specificity and *p* = 0.025 for accuracy). Differences in sensitivity were not statistically significant (*p* > 0.45 for all). The radiologist’s score was further evaluated when supported by CAD systems. In both cases, his sensitivity score increased significantly (for S-Detect, 1: 91.9%, *p* = 0.031; for S-Detect 2,: 93.0%, *p* = 0.016). However, and interestingly, his scores for specificity (for S-Detect 1, 81.1%; for S-Detect 2, 67.4%; *p* < 0.001 for both) and accuracy (for S-Detect 1, 85.3%, *p* = 0.023; for S-Detect 2, 77.5%, *p* < 0.001) decreased [38]. Other studies have evaluated the same S-Detect program [32,33,39]. Wei at al. compared CAD of a CNN-based system to the works of four radiologists: radiologist 1 (one year of experience), radiologist 2 (four years of experience), radiologist 3 (nine years of experience), and radiologist 4 (twenty years of experience). They analyzed 204 nodules from 181 patients. Only radiologist 4 achieved higher specificity, accuracy, and AUC than S-Detect (respectively, 75.0% vs. 65.2%, *p* = 0.052; 84.8% vs. 77.0%, *p* = 0.010; and 0.859 vs. 0.782, *p* = 0.005). Compared with the other radiologists, AI demonstrated higher performance. In the next step, the radiologists revised their diagnoses on the basis of the AI assessment; it was found that the less experienced radiologists made a higher number of revisions: 31, 25, 9, and 5, respectively. CAD contributed to a significant increase in the performance of only radiologists 1 and 2 (radiologist 1: specificity 37.5% vs. 58.9%, accuracy 63.7% vs. 75.0%, and AUC 0.666 vs. 0.767; radiologist 2: specificity 49.1% vs. 59.8%, accuracy 65.2% vs. 74.5%, and AUC 0.669 vs. 0.761; *p* ≤ 0.002 for all). In radiologists with more experience (radiologists 3 and 4), no increase was observed or the increase was statistically insignificant (*p* > 0.05) [32].

Comparisons of the effectiveness of CNNs and non-neural network algorithms are frequently encountered in the literature (Table 2). Ouyang et al. compared CNNs with the algorithms of four non-neural networks: random forest, SVM, kernel nearest neighbor, and naive Bayes. For all, the training set consisted of 700 nodules and the validation set 479 nodules. The performance of all five algorithms was better than that of two experienced radiologists with 15 and 17 years of experience. AUC values decreased in the following sequence: 0.954 for SVM and random forest, 0.940 for naive Bayes, 0.937 for kernel nearest neighbor, 0.928 for CNN, and 0.830 for the radiologists. Among the AIs mentioned above, the CNN-based program had the weakest results [40]. In the previously mentioned study by Kim et al., the SVM program (S-Detect 1) obtained higher specificity and accuracy than the CNN (S-Detect 2): 82.6% vs. 68.2%, *p* = 0.004 and 81.7% vs. 73.4%, *p* = 0.025. The CNN showed higher sensitivity (81.4% vs. 80.2%), although this difference was not statistically significant (*p* > 0.999) [38].

Park et al. compared the results of CNN, SVM, and radiologists. The radiologists were further divided into senior (5–20 years of experience) and junior (1–2 years of experience) groups. In all, 4919 nodules were used to train the deep neural networks. All groups were then validated on a sample of 286 nodules in 265 patients, with the senior radiologists examining 184 nodules and junior radiologists examining 102 nodules. Overall, CNN performed better than SVM, with sensitivity 91.0% vs. 90.4% (*p* not stated), specificity 80.8% vs. 58.5% (*p* < 0.001), and accuracy 86.0% vs. 75.9% (*p* < 0.001), respectively. When comparing the performance of AI to that of radiologists by experience, it was observed that (1) experienced radiologists showed better performance than less experienced radiologists and (2) there was more overlap between AI performance and senior radiologists’ CNN performance than AI performance and senior radiologists’ SVM performance. Comparing experienced radiologists, CNN, and the SVM on a group of 184 nodules, the results were as follows: sensitivity 92.9% vs. 90.8% vs. 90.8%, specificity 87.2% vs. 84.9% vs. 58.1%, and accuracy 90.8% vs. 88.0% vs. 75.5%, respectively. For specificity and accuracy, comparisons of experienced radiologists with SVM and CNN with SVM yielded *p* < 0.001. The results of comparisons of experienced radiologists with CNNs were not statistically significant, with *p* > 0.25 for both. Less experienced radiologists showed significantly poorer overall performance on their group of 102 nodules than did senior radiologists on theirs, with similar AI scores in both groups (junior radiologists: sensitivity 96.6%, specificity 56.8%, and accuracy 79.4%). The fact that a relatively large sample was represented by malignant nodules and the majority of them were papillary thyroid carcinomas could be considered the main limitation of this study [41].

Instead of analyzing the results of AI based on different mechanisms of action separately, as Ouyang et al. [40] and Park et al. [41] did, Nguyen et al. proposed the cooperation of both types of algorithms in clinical practice to exploit the potential of each. A special procedure algorithm was created. First, a non-neural handcrafted method algorithm was used, in this case fast Fourier transform (FFT). The program classified the US image on the basis of a specially developed point scale. If the score was unambiguous, that was the end of the diagnostic process, and the image was classified as either benign or malign. If the algorithm could not conclusively resolve the nature of the lesion, then the same image was independently evaluated by two different CNN algorithms, resulting in two different scoring results. The final score was taken as one of three: MIN (the lower score of the two proposed), MAX (the higher score of the two), and SUM (the arithmetic average of the two scores). This idea is based on the fact that in cases that are not in doubt the program “taught the picture” by a human expert will perform better. In unclear cases where the clinician is unsure of the diagnosis, the AI is allowed to learn the image on its own, allowing it to see patterns that are elusive to the human eye, and can make a correct assessment in such cases. After this CAD system was developed, its usability was tested against the available Thyroid Digital Image Database. A total of 237 images were used to train the CNN, and 61 images were used to evaluate the CAD system. Initially, both CNNs were evaluated individually without combining their evaluation with the FFT algorithm, and both obtained similar results in sensitivity (81.8% and 84.0%), specificity (72.5% and 74.0%), and accuracy (80.8% and 82.4%). Next, the performance of the combined CNNs (hereafter without FFT) was checked on the basis of MIN, MAX, and SUM. The highest sensitivity, specificity, and accuracy values were achieved for the MAX method, with 95.1%, 78.7%, and 91.2%, respectively. Finally, the performance of the proposed method was checked in combination with FFT and the CNN. The highest sensitivity (96.1%) and accuracy (92.1%) were achieved for the MAX method, while the highest specificity (77.2%) was achieved for the MIN method (vs. 65.7% for the MAX method) [42].

Sun et al. focused on combining CNNs with non-neural networks as well, though in a slightly different form. They proposed a CAD that can extract features from US images using both CNN (VGG-F) and handcrafted methods (histogram of oriented gradient, local binary patterns, and scale-invariant feature transform). The extracted features are then classified by the SVM algorithm. The entire experiment consisted of two stages. Initially, the researchers compared the performance of the AI itself on the basis of the method used to extract features from images. The fused method (VGG-F-based) scored better than the VGG-F method and the handcrafted method conducted separately (sensitivity 94.3% vs. 92.8% vs. 88.9%, specificity 91.1% vs. 89.2% vs. 84.7%, accuracy 92.9% vs. 91.4% vs. 87.5%, and AUROC 0.959 vs. 0.932 vs. 0.904, respectively). The second part compared the performance of CAD and radiologists with 6–10 years of experience on 550 thyroid nodules. The CAD system scored better than the radiologists (sensitivity 96.4% vs. 93.1%, specificity 83.1% vs. 67.2%, and accuracy 92.5% vs. 87.1%, respectively). The AUC for CAD was significantly higher than that for radiologists, at 0.881 vs. 0.819, *p* = 0.0003 [43].

Thomas et al. proposed a program (AIBx) based on a different principle than the others. AIBx is an image similarity algorithm, which contains in its memory photos previously classified by human experts. Its operation is based on comparing new photos to those in the initial dataset and selecting the most similar one. It then returns the same value for the new photo as the photo from the dataset classified by the human expert. The performance of AIBx was compared to that of a CNN (ResNet 34) trained on a database of 2025 images from 482 nodules. The test set consisted of 103 images from 103 nodules. AIBx achieved better performance than the CNN, with sensitivity 87.8% vs. 84.8%, specificity 78.5% vs. 74.3%, and accuracy 81.5% vs. 77.7%, respectively. The *p*-values were not reported [44]. Swan et al. externally validated the utility of AIBx. Performance based on 257 nodules from 209 patients was worse than in the original study, with sensitivity 78.4%, specificity 44.2%, and accuracy 51.0%. As the authors of the paper themselves note, the different result of their validation may be due to differences in the features of the images used in the test sets of the two studies rather than imperfections in the algorithm itself. The Thomas et al. study used US images from the same model of camera for the training set and the test set, while Swan et al. used an off-the-shelf program (trained on images from the US model of Thomas et al.) to evaluate images derived from a different US model, which was used to create the test set in the Swan et al. paper. Both ultrasound machines produced images of thyroid nodules that differed in texture and size, which may have significantly affected the results of the studies [45].

In summary, studies in recent years suggest significant benefits of using CAD systems in diagnosing thyroid nodules, particularly for less experienced radiologists [29,30,32,33]. AI assistance for further diagnostic decisions could contribute to a significant reduction in unnecessary FNAB [30,34]. The benefits of AI in assisting more experienced clinicians are not as clear [32,38]. CADs based on CNNs seem to perform better than programs based on non-neural networks [24,41], although this needs to be resolved through further research, as researchers are not unanimous [38,40]. Combining the two technologies for even more accurate results seems to be an interesting idea [42,43]. New solutions are being sought from a programming perspective, for example, AIBx [44]. It should be borne in mind that the algorithms, especially CNNs, can differ significantly from one another from a programming perspective and that each is trained on different datasets consisting of images varying in number and quality. Consequently, the obtained results may differ between studies, and this should be kept in mind when deciding to adapt AI systems for personal use in clinical practice. It is worth noting that in most of the mentioned studies the main limitation was a relatively large sample represented by malignant nodules (most of them being PTC), which may have influenced the results.

### 3.2. Cytopathology

FNAB is a standard procedure for diagnosing thyroid nodules, and is performed on the basis of US indications and clinical data on the patient and nodules [46,47,48]. The results of cytological examination can then help to decide what type of treatment should be used [49,50]. As such, it is a diagnostically significant test, which has led to many studies for its improvement through the use of AI.

Guan et al. used AI to distinguish PTCs from benign thyroid nodules on the basis of cytological images taken with FNAB. They used two algorithms, VGC-16 and Inception-v3, both of them based on the deep convolutional neural network. They used nodule samples from 279 patients, from which 887 images were generated. They used 407 images obtained from PTC patients and 352 images from benign nodule patients as the training set and 69 images from PTC patients and 59 images from benign nodule patients as the test set. To establish the ground truth, they used the histopathological results in the case of PTC patients and clinical data and laboratory and imaging results in the case of patients with benign nodules. Sensitivity and specificity in detecting PTCs among cytology preparations with PTC and benign nodule images were 100% and 94.91% for VGC-16 and 98.55% and 86.44% for Inception-v3, respectively [51]. These are high results. However, this work did not compare the performance of the algorithms with that of humans; importantly, only preparations of patients with PTCs and benign nodules were included in the study. Although this study does not prove the practical application of the algorithm in clinical settings, it does indicate future prospects in developing the algorithm with AI capabilities.

A similar topic was researched by Sanyal et al., who studied the effectiveness of CNNs in distinguishing papillary thyroid carcinoma and nonpapillary thyroid carcinoma. For this purpose, they used microphotographs of regions of interest from thyroid smears taken by fine needle aspiration cytology. They used 184 PTCs and 186 non-PTCs images at magnification x40 or x10 taken from 20 smears of 20 patients as a training set. As a test set, 42 microphotographs of PTCs and 132 microphotographs of non-PTCs were used, of which the images were of the same fragment of the smear, with one at x40 magnification and the other at x10. The performance of the algorithm was evaluated using two criteria: first, that the lesion must be marked as papillary thyroid carcinoma in any of the magnifications (×10 or ×40), and second, that the lesion must be marked as papillary thyroid carcinoma at both ×10 and ×40 magnification to be diagnosed as papillary thyroid carcinoma. The results were, respectively, sensitivity 90.48 and 33.33, specificity 83.33 and 98.48, and diagnostic accuracy 85.06 and 82.76. This creates a good perspective on the classification of PTCs, as they are the most common malignant neoplasm of the thyroid [52]. Unfortunately, the CNN results were not compared to what a human would achieve.

Elliott et al. presented an interesting use of AI. They used a CNN first to detect regions of interest (ROIs), defined as follicular groups, on whole-slide images (WSIs) taken by FNAB as a basis to then determine the TBSRTC category and the final pathology (benign/malignant). In all, 799 WSIs were used as the training set and 109 WSIs as the test set. The area under the ROC achieved by AI in differentiating ROIs from non-ROIs was 0.985. The sensitivity and the specificity in recognizing the malignancy of the final surgical pathology were 92.0% and 90.5%, respectively. The AUCs in the accuracy of the diagnoses made by AI and pathologists were 0.932 and 0.931, respectively. Interestingly, it was noted that AI was worse at determining the malignancy of thyroid nodules for those nodules in the benign and malignant cases categorized according to TBSRTC. Therefore, the operation of the algorithm was modified such that a pathologist’s assessment was retained in cases when categorization of the nodule diagnosis as benign or malignant was made according to TBSRTC. For the other categories, however, AI was used to determine the malignancy of the nodules themselves. Thus, the specificity of AI and humans together in determining the malignancy of the thyroid nodules increased from 90.5% to 92.9% and the AUC from 0.931 to 0.962. This was a better result than that achieved by either AI or pathologists alone [53].

Two years after their previous paper was published, some members of Elliott’s team focused on better understanding the part of this algorithm responsible for detecting ROIs. Working on the same database of the patients, Dov et al. assessed how this algorithm may be able to make finding ROIs easier for a cytopathologist in a different way by examining a thyroid slide presented as a WSI. Microscopic analysis of a slide taken through FNAB is often hampered and slowed by artifacts such as blood, serum, or empty space. This algorithm was designed to search for fragments of the slide from which a diagnosis could be made, thereby saving the subject time and energy and allowing them to focus on the relevant fragments. The algorithm was able to analyze the entire WSI of the thyroid and return 100 ROI images from it containing groups of follicular cells. Significantly, the 100 ROI images represented only about 0.2% of the area on the slide. These 100 ROI images per WSI were then presented to the experienced cytopathologist, who, on this basis alone, was tasked with assigning the nodule the appropriate category based on the Bethesda System for the Reporting of Thyroid Cytopathology (TBSRTC) and determining whether it was benign or malignant. The performance of this algorithm was then compared with the performance of the same cytopathologist who had performed the same task based on the WSIs 117 days earlier. The final surgical pathology result was considered as the ground truth of each WSI. On the basis of pairwise weighted ĸ statistics, the concordance of the cytopathologist’s evaluation of the specimen using only the WSIs and using only the ROIs indicated by the AI in the assignment to TBSRTC categories and in the risk assessment was calculated as ĸ = 0.924 and 0.834, respectively. The concordance of the cytopathologist’s evaluation of the specimen using only the WSIs compared with the ground truth was ĸ = 0.845 and 0.669, respectively. Unfortunately, there was no analogous comparison of evaluation based on the ROIs and ground truth. On the basis of the high concordance between the cytopathologist’s assessment using WSIs and ROIs and the cytopathologist’s assessment using WSIs and the ground truth, the researchers concluded that their algorithm could realistically help in accurately evaluating thyroid nodule biopsy samples [54]. The above works present the significant achievements of AI in helping to classify thyroid nodules. The ability to distinguish PTC from benign thyroid nodules or non-PTC thyroid carcinoma is an important achievement, as PTC is the most common thyroid cancer. The ability to distinguish PTC from a benign lesion is therefore a very important step in the development of AI for the classification and diagnosis of thyroid nodules. In addition, AI’s contribution to the detection of ROIs provides an aid to histopathologists in evaluating cytopathologic smears. This can save time for diagnosticians, and thereby reduce diagnostic costs.

### 3.3. Whole-Slide Imaging of Frozen Sections 

Pathological diagnosis is now the cornerstone of the diagnosis of an excised lesion [55]. One helpful method for the intraoperative diagnosis of thyroid nodules is the intraoperative frozen section (FS) [56,57,58,59,60]. Although there is a great deal of controversy about the utility of this method, and it is discouraged in many cases [61,62,63,64,65], research involving its improvement with AI is underway and could increase the importance of this method in the future.

Li et al. presented an algorithm by which they assigned thyroid nodules from intraoperative FS samples into benign, uncertain, or malignant categories. The multistage algorithm focused on first detecting an area of tissue from an image and then dividing it into patches and assigning the patches to one of these categories. Then, on the basis of the classification of patches, i.e., the number of each class, the entire sample was categorized as benign, uncertain, or malignant. An effort was made to apply sufficiently sharp criteria for classification into the malignant and benign groups in order to not suggest unnecessary resections (or vice versa) and to classify uncertain cases into the uncertain group. In all, 349 FSs were used as the training set and 259 FS samples as the test set. The combined accuracy for all categories was 216/259 (83.4%). Sensitivity and precision in detecting benign, uncertain, and malignant categories were 71.8% and 95.3%, 100% and 16.7%, and 88.6% and 96.7%, respectively. In addition, the typical whole-slide image was diagnosed within 1 min. Although this is not a result that allows the algorithm to replace a pathologist, the authors suggested its possible use in centers with a lack of qualified pathologists and that the use of such algorithms could sensitize pathologists to suspicious preparations [66].

Zhu et al. addressed a similar issue. Their main goal was to create an algorithm to categorize a tumor as malignant, benign, or rare on the basis of the WSIs of FSs from thyroid lesions and to then recommend the investigation of rare tumors to pathologists, the reason being the difficulty that AI has in categorizing rare tumors because of the small number of samples that can be used for the training set and because of the complexity of these preparations. The researchers used WSI slides of 200 and 53 PTCs and 296 and 61 nodular goiters as training and validation sets, respectively. In other words, they used only the slides of tumors they defined as common. As Testset1, they used WSI preparations of 283 PTCs and 334 nodular goiters. As Testset2, they used the same preparations as in Testset1, with the addition of 147 WSI preparations they defined as rare, which included five other thyroid cancers, 72 thyroid adenomatous lesions, 45 thyroid fibrous calcified nodules, and 25 other benign thyroid lesions. The AUC for classifying specimens as malignant from all cases in Testset1 by patch-UNet, which was trained on a training set basis, was 0.986. When the same algorithm was applied to detect malignant specimens from all cases in Testset2, the AUC dropped significantly, to 0.946, indicating a significant deterioration in the performance of the algorithm in classifying rare cases and pointing to the need for rare cases to be selected for evaluation by a pathologist rather than the algorithm. Thus, a three-stage classification based on the decision tree and patch-UNet models was created to select rare tumors and designate them for analysis by pathologists, with the remaining cases designated as benign or malignant. The sensitivity for determining a rare category was 0.882 and the precision was 0.498, resulting in 255 of the 764 WSIs being referred for pathologist evaluation. Of the remaining 509 WSIs, only eight were misclassified. After comparing the efficiency of classification by the algorithm with that by the pathologist, *p* > 0.05 was achieved, indicating that there was no statistically significant difference. This is an interesting solution, as it allows benign and malignant cases to be identified with human-like accuracy with the use of the AI uncertain cases sent back for human evaluation. If refined, this algorithm could be an important aid in the diagnosis of thyroid nodules [67].

Chen et al. presented the use of CNN algorithms for classifying WSIs of FSs of thyroid nodules. For this purpose, a pathologist first viewed an WSI and then marked the ROIs to be analyzed by AI for classification as malignant, uncertain, or benign. The pathologist marked the ROIs rather than the AI in order to reduce the risk of misdiagnosis due to misidentification of the ROIs. ROIs that had previously been classified by two pathologists were used as the training set. To test the effectiveness of the AI, five-fold cross-validation was used. The total dataset consisted of 345 WSIs, from which 671 ROIs were obtained. The cross-validated classification accuracy was 96.1%. Moreover, in addition to classifying thyroid nodules, the developed algorithm was designed to return images of ROIs to the examining pathologist that were similar to those in the sample. This is a useful idea, as the examining pathologist would have similar images to those they see, allowing them to compare the current finding to the previous ones. This could significantly facilitate diagnosis, especially for novice pathologists [68].

An alternative application of AI in the intraoperative diagnosis of thyroid nodules is the translation of FSs of WSIs to virtual formalin-fixed and paraffin-embedded (FFPE) sections. Although the validity of using FS for the intraoperative diagnosis of thyroid nodules is often questioned due to its low efficacy [69,70,71,72,73], an FFPE section is considered to provide a better picture of the tissue than an FS WSI. However, it is not used for intraoperative diagnosis due to the excessive time required to create the specimen. Therefore, Siller et al. created an AI algorithm to translate virtual frozen sections into virtual paraffin sections. On the basis of 80 slides taken from 40 patients with PTC or follicular thyroid carcinoma, from which 40 FSs and FFPE sections were each taken, they created three different algorithms based on generative adversarial networks designed to convert images from FS to FFPE sections. In Experiment 1, the six pathologists were then tasked with evaluating whether the created images were better than the original images from FS; in Experiment 2, they were tasked with being able to distinguish the original FFPE sections from those converted from FS. In Experiment 1, the processed FS sections scored an average of 0.6. This means that, as per the pathologists, 60% of the AI-processed images were of better quality than the originals from FS. In Experiment 2, depending on the algorithm, experts indicated the original FFPE images with an average efficiency of 62% to 97%. The average efficiency for the three algorithms was 79%, where 50% would indicate random selection and 100% would always select the original image. Unfortunately, the impact of these image transformations on intraoperative decision-making or patient outcomes was not evaluated. However, this study offers the interesting prospect of improving the quality of FS images with the goal of improved intraoperative diagnosis of thyroid nodules, and, thereby making better treatment decisions [74]. The above papers demonstrate the practical use of AI in the diagnosis of thyroid nodules. The results of AI in determining the malignancy of nodules intraoperatively does not yet allow for the replacement of diagnosticians; however, it certainly holds the promise of helping them in making diagnoses in the future [66]. An interesting solution was presented in the paper by Zhu et al., where the authors proposed using AI in the classification of more clear cases and sending the more severe ones for evaluation by a histopathologist [67]. This may be the first step in introducing AI for practical use in the diagnosis of thyroid nodules in the future. The AI function presented by Chen et al. for returning slide-like images of other slides for diagnosis seems to be an interesting option as well [68]. This could be an aid to novice diagnosticians in the future, as could the algorithm presented by Siller et al. for converting FS images into FFPE virtual sections [74]. The presented algorithms create many interesting options for future use, from facilitating doctors’ diagnosis to performing it almost entirely on their own.

### 3.4. Probe Electrospray Ionization Tandem Mass Spectrometry (PESI–MS)

One diagnostic method for thyroid nodules that has recently been studied is mass spectrometry. There are many papers indicating the potential of this method in determining the nature of thyroid nodules [75,76,77,78].

Wang et al. proposed determining the malignancy of thyroid nodules by using artificial intelligence to analyze the PESI–MS results of FNAB samples. On the basis of US results, presence of the BRAF gene, and cytopathology, patient cancers were defined as malignant (98), benign (110), or undetermined (42). Three algorithms were then created which were trained and tested exclusively on the basis of 208 patients previously defined as having malignant or benign cancers, with a training-set-to-test-set ratio of 8:2. On the basis of extensive analysis, nine components were identified as being of major importance in determining the malignancy of a nodule. The best performance among the three algorithms was attained by the multilayer perceptron method, which was able to determine the malignancy of the nodule with a sensitivity of 88.9% and a specificity of 95.7%. This method was then used to determine malignancy in patients previously identified as having cancer of an undetermined variety and on 17 new patients, with the difference that all 208 patients from the earlier training set and the test set were used as the training set this time. Of the 37 patients with undetermined cancers (five patients lost contact with the researchers during the six-month follow-up) and 17 new patients, the algorithm was able to correctly determine the malignancy of the samples with an accuracy of 72.7% and 82.4%, respectively. Considering the low time required with this method to determine the malignancy of the sample (only 10 min), this is a promising result that could be improved with a better algorithm or FNAB technique [50]. Table 3 presents a comparison of microscopic and mass spectroscopy methods in the classification of thyroid nodules, especially in differentiating malignant nodules from others.

### 3.5. Nuclear Medicine

When diagnosing thyroid nodules, in addition to simply detecting them and determining their morphology, it is important to assess their secretory status. For this purpose, laboratory tests are usually performed, especially the measurement of thyroid-stimulating hormone. However, when production of serum thyroid-stimulating hormone is suppressed, nuclear medicine techniques [79], which are used to assess thyroid function, become crucial [80]. Certain benign thyroid pathologies, including Graves’ disease and Hashimoto thyroiditis, may predispose a person to nodule formation [81,82]. Tests such as scintigraphy and thyroid single-photon emission computed tomography (SPECT) can help to differentiate the etiology of such nodules [83,84].

Yang et al. created an AI algorithm to automatically classify thyroid scintigrams on the basis of fitting one of four 99mTc-pertechnetate uptake patterns. Four deep CNNs were constructed to distinguish between the following uptake patterns: diffusely increased, diffusely decreased, locally increased, and heterogeneous uptake. The overall accuracy of the trained models was above 90%. As an adjunctive technique for identifying the cause of thyrotoxicosis, 99mTc-pertechnetate thyroid scintigraphy may be particularly useful for distinguishing Graves’ disease from toxic multinodular goiter. Additional AI assistance could help in diagnosing thyrotoxicosis easier faster as well as enabling physicians to more consistently interpret thyroid scintigrams. However, the created model has a number of limitations. The accumulation of minor differences in images acquired from different institutions and from different devices can affect the final diagnosis by AI. In addition, the differentiation of the “heterogeneous uptake” pattern from the “diffusely increased’ pattern remained imperfect. While the described AI system could be of great help in diagnosing thyroid nodules, it needs improvement [85,86].

Qiao et al. constructed three deep CNN models and compared their performance in interpreting thyroid scintigrams with that of first- and third-year nuclear medicine residents. The images were classified as showing Graves’ disease, subacute thyroiditis, or no disease. Thus, the described model lacked the ability to identify other thyroid pathologies, particularly nodules. However, it was shown that the created system could be a significant diagnostic aid in the described cases. The diagnostic performance of all three models exceeded that of the first-year residents. Furthermore, there is a real prospect of improving the described model. Supplementing it with images of other thyroid diseases has the potential to create a useful diagnostic aid, especially for younger and less experienced physicians [87].

In their retrospective study, Currie et al. compared the effectiveness of scintigraphy with that of biochemical tests in the context of diagnosing hyperthyroidism. In the analysis of scintigrams, they used constructed artificial neural network (ANN) and CNN models, which provided an accuracy of 84.6% and 80.5%, respectively. The study showed that scintigraphy is a useful method for identifying patients with hyperthyroidism. Machine learning and DL models can be useful as physician-assisted second reading systems to improve the accuracy of diagnosis [88]. Such an approach can help in the diagnosis of thyroid nodules by determining their secretory status.

Medhus et al. studied the possibility of using CNNs as an aid in evaluating thyroid scintigrams as well. The model they created automatically classifies images as having a “detectable hypofunctioning lesion” or having “no detectable hypofunctioning lesion”. However, their study included patients with other pathologies, including thyroid nodules. No significant difference was found between the accuracy of the AI system and that of experienced nuclear physicians in detecting hypofunctional lesions [89].

Ma et al. constructed a CNN model that establishes a diagnosis based on SPECT images. Because this technique assesses the function of the gland as well, the following possible diagnoses were considered: Graves’ disease, Hashimoto’s disease, subacute thyroiditis, and normal. The effectiveness of the system was compared with that of other CNN methods. The results of the modified DenseNet network architecture and the improved training method proposed in the described work were the most promising. This result suggests prospects for the further development of DL methods in the context of their use in diagnosing thyroid nodules, and it may be possible to use them widely in future [80,90,91,92]. Ma et al. constructed another CNN model to diagnose thyroid disease based on SPECT. The system they developed was designed to distinguish between four types of thyroid disease: hyperthyroidism, hypothyroidism, methylene inflammation, and Hashimoto’s disease. The method developed with an enhanced structure for DL achieved high diagnostic accuracy [92,93]. The aforementioned papers show that properly trained CNN models based on SPECT can be used to differentiate a diverse spectrum of functional thyroid pathologies. Further work on AI algorithms is needed, as it has been shown that there is room for improvement in their accuracy.

The publications described here clearly demonstrate that AI techniques can bring numerous benefits when introduced into the diagnosis of thyroid nodules by nuclear medicine. These include minimizing the risk of misdiagnosis, facilitating the work of nuclear physicians (especially those with less experience), and speeding up diagnosis, which can lead to associated reductions in costs. Although the aforementioned methods require refinement before they can be introduced into routine use, the prospects remain promising. Table 4 summarizes the information presented in the foregoing section.

### 3.6. Optimization of the Diagnosis Process

The basis of good diagnostics is selecting the right method for a particular case. However, conducting as many tests as possible is not an optimal solution, as it can be costly and have a negative impact on patient well-being in terms of both mental and physical health [94,95]. Islam et al. constructed a DL-based system that uses data extracted from electronic health records to select the appropriate laboratory test [96]. This work is not particularly about thyroid illnesses diagnosis; rather, it is about the use of AI to more generally optimize laboratory diagnostics, which could be a good perspective for future papers that focus only on patients with thyroid problems. Over-testing is a real problem afflicting every branch of medicine [97], including thyroid function testing [98]. Selecting the right test for a particular patient is a difficult task that requires analysis of numerous variables. In this regard, the assistance of AI could result in great convenience for both physicians and patients. AI systems can help reduce the number of unnecessary tests, which would bring many benefits.

### 3.7. Related Works

While studying the literature on the use of AI in the diagnosis of thyroid nodules, we came across a great many papers addressing this issue to some extent. Most of the extant publications are on ultrasound diagnostics. Finding articles on the application of AI to diagnosis using other techniques required a more in-depth search. There are few papers that concisely summarize the latest developments in the development of diagnostics using AI. For example, while there are reviews that focus exclusively on ultrasound [99,100], or cytopathology [101], our paper is not limited to one particular diagnostic method. It describes the novelties and prospects of the broad spectrum of available methods for the diagnosis of thyroid nodules, including ultrasonography, cytopathology, histopathology, and nuclear medicine techniques. Several publications only provide information on the diagnosis of thyroid cancer [102]. In this paper, we address the diagnosis and classification of nodules of various etiologies, including noncancerous ones. Sorrenti et al. have published a paper with similar topics to ours; however, they analyzed articles from a much broader time spectrum (from 2012–2022), while our work focuses only on recent developments from 2018 to date. Moreover, in our paper the issues are grouped by type of diagnostic method, while Sorrenti et al. divided their work by AI techniques [103]. We wanted the issues presented in our work to be easy to analyze for their future application in clinical settings. We believe that our publication, which groups together only the latest developments in all the useful methods of diagnosing thyroid nodules of various backgrounds, can be of value for doctors and researchers.

## 4. Conclusions

Our review of the recent literature suggests that AI may find application at various stages in the diagnosis of thyroid nodules. These include US, cytopathological or histopathological studies, and nuclear medicine techniques. According to research reports in recent years, AI seems to match experienced radiologists in US evaluation in terms of accuracy. Thus, it can be successfully used in CAD, a solution that would be particularly helpful for less experienced physicians. There would be significant benefits for patients, including the reduction of unnecessary FNABs. Future investigation is needed to assess the accuracy of combining two methods, including CNN and non-neural network algorithms. Furthermore, from a programming perspective, new solutions remain necessary. There is a prospect of using AI systems to evaluate tissue materials in both preoperative and intraoperative diagnosis. Numerous models have been developed that can effectively distinguish whether a nodule is malignant or benign on the basis of images of slides. However, these techniques require refinement. In the future, however, they could provide invaluable assistance when treatment decisions are being made, especially decisions regarding the degree of radicality of surgery. AI may find applications in diagnostics using nuclear medicine methods. A number of publications have demonstrated the effectiveness of AI in evaluating thyroid scintigrams and SPECT images. With the assistance of AI, it would be faster and easier to evaluate the secretory status of nodules, and there would be a lower risk of confusion. The limitation of many of the reviewed works was a poor spectrum of differentiated thyroid pathologies, and future investigations remain needed to extend the utility of the described methods. AI techniques can help in informing therapeutic decisions as well as in making diagnostic decisions, which could reduce the number of unnecessary tests, thereby reducing costs. We have listed numerous advantages of different AI methods; however, it is important to remember the risks as well. These include the unsolved problem of legal responsibility. For example, treatment may be delayed and prognosis may become worse in situations where a malignant thyroid nodule is not diagnosed. In addition, a false diagnosis of a badly progressing disease can significantly affect the patient’s mental state. Who would be responsible for the mistakes made by AI? The other issue that may discourage the clinicians from introducing the described methods into their routine is the associated danger of job loss. Before any of these techniques are accepted, all of the pros and cons need to be analyzed. In addition, the presented work has several limitations. The search for relevant publications was limited to two platforms, Google Scholar and PubMed, and specific phrases were used for research, meaning that there may be a large number of papers that should have been mentioned but were omitted. In addition, we were limited to reviewing papers in English. Furthermore, several of the methods cited require further research before they can be introduced into routine diagnostics. Nonetheless, we believe that our paper clearly presents the latest reports on the use of AI in the diagnosis of thyroid nodules, and that it represents a useful resource for both specialists and learners. By presenting the latest developments in the field of thyroid nodule diagnosis, we aim to spread interest in using AI for this purpose. Table 5 presents a list of papers used in our manuscript along with a short summary of each.

## Figures and Tables

**Table 1 cancers-15-00708-t001:** Diagnostic performance of CNN-based AI and radiologists.

Group	Sensitivity	Specificity	Accuracy	AUC
Junior radiologist ≤ 4y	64–96%	38–84%	64–83%	0.67–0.82
Senior radiologist > 8y	75–97%	63–96%	73–94%	0.73–0.86
AI system (CNN)	74–95%	65–94%	73–94%	0.78–0.94
CNN-assisted junior ≤ 4y radiologist	87–95%	59–81%	75–87%	0.76–0.87

AUC—an area under the receiver operating characteristic curve; AI—artificial intelligence; CNN—convolutional neural network.

**Table 2 cancers-15-00708-t002:** Comparison of the performance of different AI algorithms.

Group	Sensitivity	Specificity	Accuracy	AUC
CNN [29,30,32,38,40,41]	74–95%	65–89%	73–89%	0.78–0.94
SVM [38,40,41]	80–90%	59–83%	76–82%	0.95
RF [40]				0.95
NB [40]				0.94
k-NN [40]				0.94
Combined 2 CNN with FFT (MAX rule) [42]	96%	66%	92%	
Combined CNN-based and handcrafted-based features extraction Methods with SVM [43]	96%	83%	93%	0.88
AIBx [44,45]	78–88%	44–79%	51–82%	

AUC—area under the receiver operating characteristic curve; CNN—convolutional neural network; SVM—support vector machine; RF—random forest; NB—naive Bayes; k-NN—kernel nearest neighbour; FFT—Fast Fourier Transform.

**Table 3 cancers-15-00708-t003:** A comparison of microscopic and mass spectroscopy methods in distinguishing the malignancy of the thyroid nodules.

Paper	Differentiation between	Sample	Set of Images		Sensitivity [%]	Specificity [%]	Other [%]
Q. Guan et al. [51]	PTC/Benign nodules	FNAB cytology	Training 759 Test 128		100	94.91	
Sanyal P. et al. [52]	PTC/non-PTC	FNAB cytology	Training 370 Test 174		90.48	83.33	Accuracy 85.06
Elliott R. et al. [53]	Malignant/Benign	FNAB cytology	Training 799 Test 109		92	90.5	
Li Y. et al. [66]	Malignant/Uncertain/Benign	FS WSI	Training 349 Test 259		Malignant 88.6 Uncertain 100 Benign 71.8		Combined classification accuracy 83,4
Zhu X. et al. [67]	Malignant/Rare/Benign	FS WSI	Training 496 Validation 114	Test1 617			AUC Malignant 98,6
				Test2 764	Rare Detection 88.2		AUC Malignant 94,6
Chen P. et al. [68]	Malignant/Uncertain/Benign	FS WSI	Totally 671				Cross-validation classification accuracy 96.1
Wang Y. et al. [50]	Malignant/Benign	PESI-MS	Totally 208 Training: Test ratio 8: 2		88.9	95.7	
			Training 208 Test 17				Accuracy 72.7
			Training 208 Test 37				Accuracy 82.4

AUC—area under the ROC curve; FNAB—fine needle aspiration biopsy; FS—frozen section; PESI-MS—probe electrospray ionization tandem mass spectrometry; PTC—papillary thyroid carcinoma; WSI—whole slide image.

**Table 4 cancers-15-00708-t004:** A comparison of works that suggest the use of AI assisted nuclear medicine methods in the diagnosis of thyroid nodules.

Paper	Nuclear Medicine Technique	AI Model	Aim of the Model	Limitations
Yang P. et al. [85,86]	99 m Tc-pertechnetate scintigraphy	Deep CNN	Classification of four patterns of thyroid scintigram: diffusely increased, diffusely decreased, locally increased, heterogeneous uptake.	Impact of minor differences in images acquired from different institutions and from different devices on the final diagnosis; imperfect differentiation of “heterogeneous uptake” pattern from “diffusely increased’ pattern.
Qiao T. et al. [87]	99 m Tc-pertechnetate scintigraphy	Deep CNN	Detection of Graves’ disease and subacute thyroiditis.	Relatively typical images of patients with Graves’ disease, subacute thyroiditis, and absence of thyroid disease were gathered to train the models; some image features regarded as suspicious were neglected and deleted from the model constructions (because of insufficient samples and class imbalances); images of more types of thyroid disease (especially thyroid nodules) need to be gathered.
Currie G. et al. [88]	99 m Tc-pertechnetate scintigraphy	ANN; CNN	Comparison of the effectiveness of scintigraphy with biochemical tests in the hyperthyroidism diagnosis.	Small group of patients (123) in the retrospective study; necessity to use an appropriately validated cutoff for the patient population.
Medhus J. et al. [89]	99 m Tc-pertechnetate scintigraphy	CNN	Detection and highlighting hypofunctioning lesions found on thyroid scintigraphy.	The impact of the quality of the images used for training on the accuracy of the model.
Ma L. et al. [80]	SPECT	CNN	Diagnosis of three categories of diseases: Graves’ disease, Hashimoto disease, subacute thyroiditis.	Insufficiently detailed classification and diagnosis of thyroid diseases (because of too little data).
Ma L. et al. [93]	SPECT	CNN	Distinguishing between four types of thyroid disease: hyperthyroidism, hypothyroidism, methylene inflammation, and Hashimoto’s disease.	Limited spectrum of distinguished thyroid pathologies.

**Table 5 cancers-15-00708-t005:** Summary of the papers used in this manuscript.

Paper	Authors	Year	Dataset	Aim
A Comparison of the Performances of an Artificial Intelligence System and Radiologists in the Ultrasound Diagnosis of Thyroid Nodules	He L.-T. et al.	2022	Training set containing 1421 images to evaluate 469 nodules in 426 patients	Evaluation of AI in diagnosing thyroid nodules and comparison with the performance of radiologists with different levels of experience
Deep Learning-Based Artificial Intelligence Model to Assist Thyroid Nodule Diagnosis and Management: A Multicentre Diagnostic Study	Peng S. et al.	2021	Training set of 18,049 images of 8339 patients to evaluate 3 different study sets: Test A (2185 images of 1424 patients), Test B (1745 images of 1048 patients), Test C (366 images and videos of 303 patients)	Development of CNN for the diagnosis of thyroid nodules and evaluation how CNN could help radiologists improve their diagnostic performance
Management of Thyroid Nodules Seen on Us Images: Deep Learning May Match Performance of Radiologists	Buda M. et al.	2019	Training set of 1278 nodules in 1139 patients to evaluate test set of 99 nodules in 91 patients	Development of AI for the diagnosis of thyroid nodules and deciding about biopsy. Comparison of AI performance and radiologists performance
The Value of S-Detect in Improving the Diagnostic Performance of Radiologists for the Differential Diagnosis of Thyroid Nodules	Wei Q. et al.	2020	Study set of 204 thyroid nodules in 181 patients	Evaluation of AI in diagnosing thyroid nodules and evaluation how CNN could help radiologists to improve their diagnostic performance
Clinical Validation of S-DetectTM Mode in Semi-Automated Ultrasound Classification of Thyroid Lesions in Surgical Office	Barczynski M. et al. [33]	2020	Study ser of 50 thyroid nodules in 50 patients	Development of CAD system for the diagnosis of thyroid nodules
Ultrasound Image Analysis Using Deep Learning Algorithm for the Diagnosis of Thyroid Nodules	Song J. et al. [34]	2019	Training set of 1358 thyroid nodules to evaluate test set of 155 thyroid nodules	Development of CAD system for predicting FNAB results of thyroid nodules
Thyroid Ultrasound Image Classification Using a Convolutional Neural Network	Zhu Y.-C. [35]	2021	Training set of 600 nodules in 592 patients to evaluate 200 nodules in 187 patients	Development of CNN algorithm for diagnosis of thyroid nodules
Ensemble Deep Learning Model for Multicenter Classification of Thyroid Nodules on Ultrasound Images	Wei X. et al. [36]	2020	Training set of 17859 images to evaluate test set of 8682 images	Development of AI algorithm for diagnosis of thyroid nodules
Computer-Aided Diagnosis for Classifying Benign versus Malignant Thyroid Nodules Based on Ultrasound Images: A Com-parison with Radiologist-Based Assessments	Chang Y. et al. [37]	2016	Test set of 59 thyroid nodules	Evaluation of CAD system for diagnosis of thyroid nodules
Real-World Performance of Computer-Aided Diagnosis System for Thyroid Nodules Using Ultrasonography	Kim H. et al. [38]	2019	Study set of 106 patients with 218 thyroid nodules	Evaluation of the diagnostic performance of CAD system for detecting thyroid cancers
A Computer-Aided Diagnosing System in the Evaluation of Thyroid Nodules-Experience in a Specialized Thyroid Center	Xia S. [39]	2019	Test set of 180 thyroid nodules in 171 patients	Evaluation of the diagnostic performance of CAD system for detecting thyroid cancers
Comparison between Linear and Nonlinear Machine-Learning Algorithms for the Classification of Thyroid Nodules	Ouyang F. et al. [40]	2019	Training set of 700 nodules to evaluate test set of 479 nodules	Comparison of the classification performance of linear and nonlinear AI algorithms for the evaluation of thyroid nodules
Diagnosis of Thyroid Nodules: Performance of a Deep Learning Convolutional Neural Network Model vs. Radiologists	Park V. Y. et al. [41]	2019	Training set of 4919 nodules to evaluate test set of 286 nodules in 265 patients	Development of deep learning-based CAD system for the diagnosis of thyroid nodules and comparing its performance with SVM-based CAD system
Ultrasound Image-Based Diagnosis of Malignant Thyroid Nodule Using Artificial Intelligence	Nguyen D. T. et al. [42]	2020	Training set of 237 images to evaluate test set of 61 images	Development of CAD system, that combine CNN and non-neural network algorithms to improve AI performance in diagnosis of thyroid nodule
Evaluation of a Deep Learning-Based Computer-Aided Diagnosis System for Distinguishing Benign from Malignant Thyroid Nodules in Ultrasound Images	Sun C. et al. [43]	2020	Training set of 1037 nodules to evaluate test set of 550 nodules	Evaluation of AI in diagnosing thyroid nodules and comparison with the performance of radiologists
AIBx, Artificial Intelligence Model to Risk Stratify Thyroid Nodules	Thomas J. et al. [44]	2020	Training set of 482 nodules to evaluate test set of 103 thyroid nodules	Development of image similarity algorithm for the diagnosis of thyroid nodules
External Validation of AIBx, an Artificial Intelligence Model for Risk Stratification, in Thyroid Nodules	Swan K. et al. [45]	2022	Test set of 257 nodules in 209 patients	External validation of AIBx algorithm
Deep convolutional neural network VGG-16 model for differential diagnosing of papillary thyroid carcinomas in cytological images: a pilot study	Q. Guan et al. [51]	2019	887 images from 279 cytological smears each from a different patient	Use of AI to differentiate PTC from benign thyroid nodules using cytological images
Artificial Intelligence in Cytopathology: A Neural Network to Identify Papillary Carcinoma on Thyroid FineNeedle Aspiration Cytology Smears	Sanyal P. et al. [52]	2018	544 images from 30 cytological smears each from a different patient	Development of ANN with the purpose of distinguishing PTC and non-PTC on microphotographs from thyroid FNAB cytology smears
Application of a Machine Learning Algorithm to Predict Malignancy in Thyroid Cytopathology	Elliott R. et al. [53]	2020	908 WSIs from 659 different patients	Development of AI algorithm to evaluate thyroid FNAB via WSIs to predict malignancy and to identify ROIs
Use of Machine Learning–Based Software for the Screening of Thyroid Cytopathology Whole Slide Images	Dov et al. [54]	2022	908 WSIs from 659 different patients	Assessing the ability of AI and screening software to identify a group of informative ROIs on thyroid FNA WSI that can be used for definitive diagnosis
Rule-based automatic diagnosis of thyroid nodules from intraoperative frozen sections using deep learning	Li Y. et al. [66]	2020	608 WSIs	Defining thyroid nodules from intraoperative frozen sections as benign, uncertain, or malignant using AI
Deep Learning-Based Recognition of Different Thyroid Cancer Categories Using Whole Frozen-Slide Images	Zhu X. et al. [67]	2022	1374 WSIs	Predicting rare categories of thyroid cancer and recommending lesion areas annotated by AI to be rereviewed by pathologists
Interactive Thyroid Whole Slide Image Diagnostic System using Deep Representation	Chen P. et al. [68]	2020	345 WSIs	Classification of frozen sections of thyroid by AI into malignant, uncertain or benign based on the suspicious regions preselected by pathologists
On the Acceptance of “Fake” Histopathology: A Study on Frozen Sections Optimized with Deep Learning	Siller et al. [74]	2022	80 WSIs from 40 different patients	Translation of virtual frozen sections into virtual paraffin sections by AI
Fast Classification of Thyroid Nodules with Ultrasound Guided-Fine Needle Biopsy Samples and Machine Learning	Wang Y. et al. [50]	2022	267 FNAB samples each from a different patient	Determining the malignancy of thyroid nodules by using artificial intelligence, analyzing the PESI–MS results of FNAB samples
Automatic differentiation of thyroid scintigram by deep convolutional neural network: a dual center study	Yang P. et al. [85]	2021	3389 thyroid scintigrams	Development of AI system that classifies the four patterns of thyroid scintigrams: diffusely increased, diffusely decreased, local increased, heterogeneous uptake
Deep Convolution Neural Network Based Articial Intelligence Improves Diagnosis of Thyroid Scintigraphy for Thyrotoxicosis: a Dual Center Study	Yang P. et al. [86]	2020	3389 thyroid scintigrams	Development of AI system that classifies the four patterns of thyroid scintigrams: diffusely increased, diffusely decreased, local increased, heterogeneous uptake
Deep learning for intelligent diagnosis in thyroid scintigraphy	Qiao T. et al. [87]	2021	1430 patients who underwent thyroid scintigraphy	Construction of three DCNN models to diagnose Graves’ disease and subacute thyroiditis by thyroid scintigraphy
Remodeling 99mTc-Pertechnetate Thyroid Uptake: Statistical, Machine Learning, and Deep Learning Approaches	Currie G. et al. [88]	2022	Thyroid scintigrams from 123 different patients	Comparison of the effectiveness of scintigraphy with biochemical tests in the context of the diagnosis of hyperthyroidism; assessment of the utility of ANN and CNN models in the analysis of thyroid scintigrams
Development of an artificial intelligence model based on the VGG19 network for automated detection of hypofunctioning lesions in thyroid scintigraphy	Medhus J. et al. [89]	2022	1724 thyroid scintigrams	Development of ANN to detect and highlight hypofunctioning lesions found on thyroid scintigraphy automatically
Thyroid diagnosis from SPECT images using convolutional neural network with optimization	Ma L. et al. [80]	2019	2888 SPECT images	Construction of CNN for the diagnosis of thyroid diseases using SPECT images
Diagnosis of Thyroid Diseases Using SPECT Images Based on Convolutional Neural Network	Ma L. et al. [93]	2018	SPECT thyroid data	Construction of CNN to distinguish four kinds of thyroid diseases: hyperthyroidism, hypothyroidism, methylene inflammation, and Hashimoto’s disease using SPECT images

## Data Availability

The datasets used and/or analyzed during the current study areavailable from the corresponding author upon reasonable request.

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
