# Peer review of "The Use of Artificial Intelligence in the Diagnosis and Classification of Thyroid Nodules: An Update"

_cancers, 2023, doi:10.3390/cancers15030708_

Round 1
Reviewer 1 Report
In this review article, the authors aim to determine the latest application of artificial intelligence in diagnosing and classifying thyroid nodules.
Structure and presentation are the main issues of this submission. References are adequate and up-to-date.
In the introduction, the authors should be more clear about the purpose and research questions being answered in this submission.
I recommend the authors discuss how they designed their review (e.g., inclusion/exclusion criteria, screening and selection of studies).
Additionally, the authors should note the structure of the review article.
In table 2, the corresponding reference for each model should also be mentioned.
It is suggested that authors create summary tables of the findings of the works described. This applies to all subsections of section 2.
I would like to know the limitations and potential issues of this study. The academic implications of this study are not reported.
Open issues for further investigation arising from this submission should be highlighted.
In general, the way of writing and presenting the article is very compressed. There are long descriptions that tire the reader.
Author Response
Dear Sir/Madam,
Thank You very much for Your feedback and guidance. They were very valuable for us and allowed us to draw attention to aspects that may have previously remained undeveloped. We have tried to take them to heart and improve our review on the basis of Your recommendations. We hope that as a result of these corrections, our review will be able to convey the subject matter of the issue to the readers of "Cancers" in the most appropriate and complete way. Below there are the comments on the revisions we have made. Thank You again for taking the time to read our manuscript.
Yours faithfully,
Maksymilian Ludwig, Bartłomiej Ludwig, Agnieszka Mikuła, Szymon Biernat, Jerzy Rudnicki and Krzysztof Kaliszewski
Reviewer 1
"In this review article, the authors aim to determine the latest application of artificial intelligence in diagnosing and classifying thyroid nodules.
Structure and presentation are the main issues of this submission. References are adequate and up-to-date."
-
- "In the introduction, the authors should be more clear about the purpose and research questions being answered in this submission."
- We are grateful for this valuable remark. In the introduction, we have developed the purpose of our work and the discussed issues.
-
- "I recommend the authors discuss how they designed their review (e.g., inclusion/exclusion criteria, screening and selection of studies)."
- We have separated the section dealing with the methodology of our review and added some information on inclusion/exclusion criteria, the number of authors making article selections and we have detailed the date range of the articles searched. We believe this will increase the quality of our review.
- "I recommend the authors discuss how they designed their review (e.g., inclusion/exclusion criteria, screening and selection of studies)."
-
- "Additionally, the authors should note the structure of the review article."
- We are really glad that You highlighted this important point. We slightly rearranged the structure of our paper. We added and expanded the methods section to make our review more organized.
- "Additionally, the authors should note the structure of the review article."
-
- "In table 2, the corresponding reference for each model should also be mentioned."
- Thank you for noticing this problem. We added missing references.
- "In table 2, the corresponding reference for each model should also be mentioned."
-
- "It is suggested that authors create summary tables of the findings of the works described. This applies to all subsections of section 2."
- Adding a table is, indeed, a good way to summarize information collected in text. We added one table to the section 3.5 (old number 2.5) in order to make the included data easier to process. Thank You for this advice.
- "It is suggested that authors create summary tables of the findings of the works described. This applies to all subsections of section 2."
-
- "I would like to know the limitations and potential issues of this study. The academic implications of this study are not reported."
- Our study has some limitations and it is, indeed, very important to mention them. We listed main limitations of our work in the conclusions, as well as we added a piece of information about academic implications, as suggested. We are really glad that You noticed that valid point.
- "I would like to know the limitations and potential issues of this study. The academic implications of this study are not reported."
-
- "Open issues for further investigation arising from this submission should be highlighted."
- Indeed, there are some issues that need more investigation. We highlighted the most important of them by mentioning them in the conclusions section. Thank You very much for this instructive suggestion.
- "Open issues for further investigation arising from this submission should be highlighted."
-
- "In general, the way of writing and presenting the article is very compressed. There are long descriptions that tire the reader."
- We understand that some descriptions may be a little too long. However, we have tried our best to describe and approximate the studies cited so that readers can best judge for themselves the quality of the studies and the information they contain. We wanted the reader's own assessment of the cited data, hence the breadth of the descriptions.
- "In general, the way of writing and presenting the article is very compressed. There are long descriptions that tire the reader."
Thank You for all these remarks. We hope that including them in our article will make our paper more interesting and legible for the readers.
Reviewer 2 Report
The manuscript reviews the application of AI in diagnosing and classifying thyroid nodules. I have the following concerns with the current manuscript:
1) Please provide the full name of those abbreviations when they are the on first emergence, e.g., AACE/ACE/AME in page 2.
2) The manuscript structure needs to be re-arranged.
3) Please provide the discussion regarding the pros and cons of AI.
Author Response
Dear Sir/Madam,
Thank You very much for Your feedback and guidance. They were very valuable for us and allowed us to draw attention to aspects that may have previously remained undeveloped. We have tried to take them to heart and improve our review on the basis of Your recommendations. We hope that as a result of these corrections, our review will be able to convey the subject matter of the issue to the readers of "Cancers" in the most appropriate and complete way. Below there are the comments on the revisions we have made. Thank You again for taking the time to read our manuscript.
Yours faithfully,
Maksymilian Ludwig, Bartłomiej Ludwig, Agnieszka Mikuła, Szymon Biernat, Jerzy Rudnicki and Krzysztof Kaliszewski
Reviewer 2:
"The manuscript reviews the application of AI in diagnosing and classifying thyroid nodules. I have the following concerns with the current manuscript:"
-
- "Please provide the full name of those abbreviations when they are the on first emergence, e.g., AACE/ACE/AME in page 2."
- We are really grateful that You spotted this mistake. We added the full name of each abbreviation that was omitted.
- "Please provide the full name of those abbreviations when they are the on first emergence, e.g., AACE/ACE/AME in page 2."
-
- "The manuscript structure needs to be re-arranged."
- Thank you for noticing this problem. W slightly rearranged the structure of our paper by adding and expanding the methods section. We hope that after these changes our publication is better organized.
- "The manuscript structure needs to be re-arranged."
-
- "Please provide the discussion regarding the pros and cons of AI"
- Thank You for this suggestion. In the conclusions section we added a brief discussion about the advantages and potential risks of introducing AI techniques into routine diagnostics.
- "Please provide the discussion regarding the pros and cons of AI"
Thank You for Your suggestions. We hope that adding this information will increase the scientific value of our article.
Reviewer 3 Report
Thank you for your work compiling this review. It is overall well-organized is a clearly stated summary of modern AI research in the field of thyroid nodules. I have only 2 suggestions/questions.
The ultrasound portion, while being a clearly stated review of the major findings of these papers, does not provide any discussion of the strengths or weaknesses of these studies that may impact the validity of the result.
I understand why the study design of Savala et al (2018) PMID: 29266871 may not fit the review criteria (although follicular adenoma versus follicular carcinoma is an interesting question that has been largely ignored in this literature). However, it is not clear why Sanyal et al (2018) PMID: 30607310 was not included.
Thank you
Author Response
Dear Sir/Madam,
Thank You very much for Your feedback and guidance. They were very valuable for us and allowed us to draw attention to aspects that may have previously remained undeveloped. We have tried to take them to heart and improve our review on the basis of Your recommendations. We hope that as a result of these corrections, our review will be able to convey the subject matter of the issue to the readers of "Cancers" in the most appropriate and complete way. Below there are the comments on the revisions we have made. Thank You again for taking the time to read our manuscript.
Yours faithfully,
Maksymilian Ludwig, Bartłomiej Ludwig, Agnieszka Mikuła, Szymon Biernat, Jerzy Rudnicki and Krzysztof Kaliszewski
"Thank you for your work compiling this review. It is overall well-organized is a clearly stated summary of modern AI research in the field of thyroid nodules. I have only 2 suggestions/questions."
-
- "The ultrasound portion, while being a clearly stated review of the major findings of these papers, does not provide any discussion of the strengths or weaknesses of these studies that may impact the validity of the result."
- We are grateful for this valuable remark. We have added this date to the work.
- "The ultrasound portion, while being a clearly stated review of the major findings of these papers, does not provide any discussion of the strengths or weaknesses of these studies that may impact the validity of the result."
-
- "I understand why the study design of Savala et al (2018) PMID: 29266871 may not fit the review criteria (although follicular adenoma versus follicular carcinoma is an interesting question that has been largely ignored in this literature). However, it is not clear why Sanyal et al (2018) PMID: 30607310 was not included"
- Indeed we didn’t include Sanyal et al (2018) in our publication. It was our unintentional omission. It’s hard to find all interesting articles when there are so many topic-relevant and topic-irrelevant articles in the databases and it’s impossible to search through the whole database to find all interesting articles. Some articles we specially omitted to not double some topics. Indeed Sanyal et al (2018) is a very interesting article and we should have included it in our research. That’s why we have added the whole paragraph about it in our review. Thank you for the suggestion of this interesting paper.
- "I understand why the study design of Savala et al (2018) PMID: 29266871 may not fit the review criteria (although follicular adenoma versus follicular carcinoma is an interesting question that has been largely ignored in this literature). However, it is not clear why Sanyal et al (2018) PMID: 30607310 was not included"
Thank You for Your suggestions. We hope that adding this information will increase the scientific value of our article.
Round 2
Reviewer 1 Report
I have no additional remarks on the revised version.
The authors have addressed my concerns.
Author Response
Dear Sir/Madame,
thank you. We're glad to hear it.
Sincerely,
Maksymilian Ludwig, Bartłomiej Ludwig, Agnieszka Mikuła, Szymon Biernat, Jerzy Rudnicki and Krzysztof Kaliszewski